# Assessment of Workplace Safety Climate among Healthcare Workers during the COVID-19 Pandemic in Low and Middle Income Countries: A Case Study of Nigeria

**DOI:** 10.3390/healthcare9060661

**Published:** 2021-06-01

**Authors:** Haruna Musa Moda, Fabian M. Dama, Christopher Nwadike, Basim S. Alatni, Solomon O. Adewoye, Henry Sawyerr, Pauline J. S. Doka, Mela Danjin

**Affiliations:** 1Department of Health Professions, Manchester Metropolitan University, Manchester M15 6BG, UK; 2Department of Urban and Regional Planning, Modibbo Adama University, Yola P.M.B 2076, Nigeria; mazafab1234@gmail.com; 3Department of Agricultural Technology, Forestry College of Forestry, Jos 930253, Nigeria; chrisnwadike@fcfjos.edu.ng; 4College of Architecture and Planning, Imam Abdulrahman Bin Faisal University, Dammam 34221, Saudi Arabia; bsalatni@iau.edu.sa; 5Department of Pure and Applied Biology, Ladoke Akintola University of Technology, Ogbomoso P.M.B 4000, Nigeria; soadewoye@lautech.edu.ng; 6Department of Environmental Health Sciences, Kwara State University, Malete P.M.B 1530, Nigeria; henry.sawyerr@kwasu.edu.ng; 7Department of Public Health, College of Nursing and Midwifery, Gombe 760251, Nigeria; paulinedoka@gmail.com (P.J.S.D.); danjin67@yahoo.com (M.D.)

**Keywords:** safety climate, safety leadership, healthcare, LMICs, Nigeria

## Abstract

The COVID-19 pandemic has presented several organizations with the opportunity to review their operational strategies, as well as the existing safety climate within their establishments. The healthcare sector is not an exception, especially those in Low and Middle Income Countries (LMICs), where most safety systems are not robust when compared with developed countries. The study aim is to assess the occupational safety climate among healthcare workers (HCWs) in LMICs using Nigeria as a case study. A cross-sectional study was adopted to measure safety climate perception among professionals working in healthcare establishment during the COVID-19 pandemic using a validated Nordic Safety Climate Questionnaire (NOSACQ-50). At the end of the survey period, 83% (433) of the responses were adjudged to have met the threshold criteria and were used to inform the study outcome. Worker safety commitment within the healthcare facilities (*M* = 3.01, *SD* = 0.42) was statistically significantly higher than management safety priority, commitment, and competence (*M* = 2.91, *SD* = 0.46), *t*(130.52), *p* < 0.001. A significant effect of the management role was found in regards to management safety priority, commitment, and competence (*F*(1, 406) = 3.99, *p* = 0.046, η^2^ = 0.010). On the contrary, the managerial position does not have a significant effect on worker safety commitment (*F*(1, 417) = 0.59, *p* = 0.440, η^2^ = 0.001). The outcome from the study showed that, where there is active promotion of a positive safety climate in healthcare sectors in LMICs, employees are more likely to engage in positive safety behaviour. To help address the identified gaps, there is the need for more effort to be made towards promoting an effective and positive safety climate across the establishment, including management and healthcare worker commitments.

## 1. Introduction

The COVID-19 pandemic has had a significant impact on existing health and safety systems in different sectors of the economy, to which the healthcare establishment has had its own share of this impact. Professionals within the healthcare sector had to find means of responding to the increasing rate of infection, as well as endure back to back overtime shifts as they dealt with the overwhelming rate of hospitalization due to the pandemic and manage other diseases and chronic conditions, all of which increase physical and mental exhaustion. In Africa and other LMICs, the states of healthcare delivery systems are mostly below the required safety standards owing to their abject state of neglect, deficiency of both human and technical resources, and the almost nonexistence of healthcare management information systems, which predate the emergence of the SARS-CoV-2 pandemic [1,2,3].

In order to warrant workplace safety performance, healthcare sectors will be required to come up with safety rules and procedures and ensure that they are applied across the organization [4]. The principle is to guarantee both employees and patients their safety, as well as the avoidance of costs associated with accident or injury. While this has been the tradition moving forward, healthcare facilities, especially in LMICs, now need to invest in the implementation of formal safety programs and risk management systems [5].

Developing a culture of safety in healthcare organizations is an important pillar, as it strives to eliminate the factors that contribute toward the management of mental and physical exhaustion, medical errors, patient harm, unsafe conditions, and the enhancement of overall patient safety [6,7,8,9]. In addition, an organization’s safety culture is an important factor that influences systems safety. Wagner, Schöne, and Rieger [10] advanced seven antecedent variables that help form a good safety climate: structural attributes of the work environment, symbolic social interaction, group and organization leadership, psychological work ownership, organizational commitment, job stress, burnout, and personality. Nieva and Sorra [11] defined safety culture as “shared attitudes, beliefs, values and assumptions that underlie how people perceive and act upon safety issues within their organizations”, whereas the term safety climate refers to the shared expression or measurable components of safety culture, such as management behaviors, safety systems, and employee safety perceptions [12]. Even though the precise denotations of safety culture and safety climate are dissimilar, these two expressions have been used interchangeably. Moving forward, the present study focus is around safety climate within healthcare settings in Nigeria.

Traditionally, safety climate has been researched in high-risk industries that include oil and gas, the construction sector, transportation, manufacturing factories, nuclear facilities, etc. [13,14,15,16,17]. However, the complex and unique characteristics of the healthcare organization further call for safety climate research to help improve both patients’ and employees’ safety. Recently, there has been growing acceptance within the healthcare establishment around the assessment of leading indicators, which are considered important in the promotion of patient safety. These factors include emphasis on production, efficiency, and cost controls, organizational and individual inability to acknowledge fallibility, and professional norms for perfectionism among healthcare providers [11,18,19]. Workplace safety climate assessment provides organization understanding of safety-related perceptions and attitudes of its workforce, and can be applied as a diagnostic tool to help identify areas within the organization that need improvement, providing needed impetus for further assessment while making use of staff input to address identified gaps [11]. As such, the benchmarking of the safety climate within healthcare facilities is now a focus, especially in developed countries that include the United Kingdom, USA, etc. [20,21].

The high increase in the number of healthcare facilities in Nigeria over the last decade has presented the challenge of the maintenance of best practices to guarantee the safety of patients and employees [3,22]. In addition, Nigeria has one of the largest pools of healthcare personnel in Africa [23]. The healthcare sector in the country makes up about one-third of the total workforce, but mostly is concentrated in urban tertiary healthcare services in the southern part of the country. Due to the present state of the healthcare services in most LMICs, healthcare workers (HCWs) perform their duties in an increasingly hazardous work environment and occupational setting [24,25,26]. Personnel in this workforce are responsible for providing quality healthcare services, even though their work places are increasingly unsafe [27], and they encounter frequent forms of hazards at work that include injuries, direct infections, stress, assault from patients and their relatives, allergies, back pain, and other musculoskeletal injuries [5,28]. The multiplier effects of occupational injuries and diseases among HCWs include economic loss, physical loss, and psychological disorders, such as stress and depression. These have an overall negative impact on the workers, their families, and the nation at large. Identifying factors relating to occupational hazards among HCWs is essential in formulating occupational health safety policy and a system that will improve productivity and the overall wellbeing of HCWs [29].

Promotion of a positive culture of safety is associated with clinical outcomes [30]; as such, the assessment of the safety climate within healthcare facilities can provide a picture of management commitment, performance, and quality of care that further promotes a positive safety culture within the organization. While there has been a resurgence in research on workplace safety within healthcare, there are limited research works that have focused explicitly on the workplace safety climate and workplace characteristics in Nigeria. The aim of the present study is to assess the occupational safety climate among healthcare workers (HCWs) in LMICs using Nigeria as a case study. The objectives of the study are to compare the impacts that individuals with managerial roles have on the seven indicators of work safety climate in the promotion of an effective safety culture in healthcare sectors; to establish the most significant determinants of the work safety climate; and to make appropriate recommendations for improving the workplace safety climate in the healthcare sector in Nigeria.

## 2. Materials and Methods

The present study measured safety climate perception among healthcare workers during the COVID-19 pandemic. The Nordic Safety Climate Questionnaire (NOSACQ-50) was used among randomly selected healthcare workers. The questionnaire consists of 50 statements, which set out to evaluate seven safety climate dimensions.

### 2.1. Sample Size

Due to the lack of official data in the public domain, an estimated 14,000 HCWs are working in the study area. To determine the sample size for the study, Fisher’s formula [31] for estimating single proportions and an estimation for minimum sample size was applied, and the estimated sample size was 423.

Fisher’s formula:$$n=\frac{Z^{2}P\left(1-P\right)}{d^{2}}$$
where:

*n* = sample size;

*Z* = standard deviation for a 95% confidence level (*Z* = 1.96);

*P* = prevalence of the attribute (50.0%);

*d* = acceptable difference (if 5%, *d* = 0.05);

*q* = 1 – *p.*

### 2.2. Study Population

At the end of the survey, 433 participants comprised of doctors, nurses, midwives, pharmacists, laboratory scientists, physiotherapist, dietitians, radiographers, community health officers, community health extension workers, environmental health professionals, etc., from five states in the northern part of Nigeria consisting of Adamawa, Gombe, Plateau, Taraba, and the federal capital territory Abuja were considered in the study. The survey was conducted using the Joint Information Systems Committee (JISC) online survey platform, and a link was sent to target healthcare management for distribution among the staff. Similarly, the survey link was sent to social media host administrators for each local professional identified, and, where deemed appropriate by the administrator, it was shared among its group members. All participants agreed to participate in the study by signing an electronic consent form before they were able to progress to the question section. The Gombe State Ministry of Health (MOH/ADM/621/1/294) granted ethics approval for the study.

### 2.3. Survey Design

The survey was conducted from October 2020 to March 2021 with different professionals working in healthcare establishments across the northeast and north central region of Nigeria. The NOSACQ-50 questionnaire was adopted to capture the employee’s perception of safety climate features that relate to supervisor and management support regarding workplace safety. A translated and validated English version of the questionnaire comprised of 50 items across seven safety climate dimensions to measure the participant’s shared safety climate perceptions was adopted. Sections of the questionnaire included: (i) management safety priority, commitment, and competence; (ii) management safety empowerment; (iii) management safety justice (six items); (iv) workers’ safety commitment; (v) workers’ safety priority and risk non-acceptance; (vi) peer safety communication, learning, and trust in safety ability; and (vii) trust in the efficacy of safety systems. Accordingly, the first three items measured the perception of safety management within the healthcare organization, while the remaining four items were related to employees’ safety commitments. Each item was rated using a four-point Likert scale of agreement (from 1 = strongly disagree to 4 = strongly agree) that corresponded to a 1–4 rating scale in case of positively formulated statements or 4–1 for the reversed statements, respectively, and attaining a high scale score was indicative of a positive response. In addition, questions on demographic characteristics were considered in the questionnaire that included age, sex, and whether the respondent held a management position [32].

### 2.4. Sampling Technique

A pilot survey was conducted among 20 HCWs to assess the instrument acceptability, validity, and reliability. The response and general comments received were found to have met the need of the study, and, as such, there was no further adjustment made to the initial questionnaire in the main study.

A cross-sectional study design was adopted for the study, and a convenient snowball sampling technique was used to reach out to the target participants. Participants were drawn using deliberate contact and sensitization of the different health professional associations and workplace units. All HCWs voluntarily completed the survey either using an online link shared on the HCW’s association internal mails or closed social media platforms or by physical administration of hard copies of the questionnaires.

### 2.5. Reliability Test

To measure the reliability for sets of latent variables in each dimension, Cronbach’s alpha test was done [33], and the result revealed that all sets of items were closely related, with an acceptable to good alpha score range of 0.703–0.810 among the seven dimensions measured. The management safety priority, commitment, and competence consisted of nine items (α = 0.787), the management safety empowerment subscale consisted of seven items (α = 0.793), safety communication, learning, and trust among co-workers’ safety competence consisted of eight items (α = 0.810), and the overall NOSACQ-50 was found to have an excellent α score of 0.932 (Table 1).

### 2.6. Statistical Analysis

The statistical data analysis was conducted using SPSS statistics version 25 (IBM, Armonk, NY, USA) after data cleaning.

Descriptive statistics using frequencies and percentages were employed to provide information on the socioeconomic characteristics of the study population. Inferential statistics carried out included a one-tailed Student’s t-test to establish statistical differences between the seven NOSACQ-50 components adopted for the study, while associations between dimensions of the safety climate were tested using the Pearson correlation. The mean scores of the safety climate were calculated for all dimensions, where a mean score of more than 3.30 indicated a good level allowing for maintaining and continuing safety developments. Lastly, a one-way ANOVA was conducted to compare the impact that employees had on the promotion of an effective safety climate. Statistical significance was set at <0.05.

## 3. Results

At the end of the survey period, 433 (80%) responses were adjudged to have met the study threshold criteria and were used to inform the study outcome. Participants’ demographic characteristics are presented in Table 2. From the results, more than half of the participants (53.7%) were male, while 40.7% identified their profession as either nurses or midwives; 45.3% of the sampled group held various managerial positions, such as director, matron, unit head, supervisor, etc. Based on the feedback generated as part of the open-ended comment at the end of the questionnaire survey, 85% of the participants said that they found the questions to be relevant and easy to understand, and had no issue with their layout.

Table 3 shows one sample t-test result conducted to determine if a statistically significant difference existed between the seven NOSACQ-50 dimensions applied in the study. From the analysis, participants’ responses on workers’ safety commitment within the healthcare facilities (*M* = 3.01, *SD* = 0.42) was statistically significantly higher than management safety priority, commitment, and competence (*M* = 2.91, *SD* = 0.46), *t*(130.52) = *p* < 0.001). In addition, dimension six (*M* = 3, SD = 0.42) was found to be statistically higher than management safety justice (*M* = 32.744, SD = 2.69), *t*(147.02) = *p* < 0.001).

To examine the extent to which scores in one dimension were related to other items in the Nordic safety climate questionnaire used, inter-correlation analyses were performed, and the results are presented in Table 4. From the results, all dimensions were statistically significant, and were found to be greater than or equal to *r* (357) = 0.33, *p* < 0.001. The management safety justice dimension was found to have a high, statistically significant correlation to management safety empowerment (*r* (357) = 0.68, *p* < 0.001) among the participants. Employees’ trust around the efficacy of existing safety systems measured was weakly correlated to management safety justice (*r* (357) = 0.33, *p* < 0.001).

A one-way ANOVA between subjects to compare the impact that employees with a managerial role had in the promotion of an effective safety climate showed a significant effect of a managerial role on the promotion of a safety climate within the healthcare facilities when compared to management safety priority, commitment, and competence (*F*(1, 406) = 3.99, *p* = 0.046, η^2^ = 0.010). On the contrary, a management role did not have a significant effect on workers’ safety commitment (*F*(1, 417) = 0.59, *p* = 0.440, η^2^ = 0.001) (Table 5). In addition, there was no significant effect on the role played by each professional group toward the promotion of a safety climate within their organization. Additionally, a low level of trust in the efficacy of existing safety systems was found among different participants in different job roles, and presented the need for improvement (*F* = 21.965, *n* = 416).

## 4. Discussion

The main purpose of the present study was to measure the safety climate among professional employees within healthcare facilities in Nigeria. The survey tool adopted (NOSACQ-50) has proven to be useful in achieving the study goal. Furthermore, the outcome from the study demonstrated the need for strengthening the safety climate within healthcare facilities, especially in LMICs. The importance of safety climate enhancement in the LMICS healthcare establishment was further demonstrated as part of the participants’ safety commitment response: (1) “We who work here try hard together to achieve a high level of safety”; (2) “We who work here take joint responsibility to ensure that the workplace is always kept tidy”; (3) “We who work here help each other to work safely and management safety priority, commitment, and competence; (1) “Management places safety before production”; and (2) “We who work here have confidence in the management’s ability to deal with safety”, where a mean score of 2.70 to 2.99 was achieved, demonstrating a fairly low level of safety practices and commitment from management. In general, the outcome shows a significant correlation between these outcomes and poor safety perceptions, low job satisfaction, and a high level of stress among the participants. Hence, there is a need for further improvement.

The relationship between safety outcomes among healthcare professionals and the organizational climate has been earlier researched [34,35]; each study concluded that the promotion of a high-quality work environment is likely to present positive effects on the workplace safety climate and outcomes. In the present study, the role played by employees identified as having a managerial position was found to have a significant effect on management safety empowerment (*F*(1, 415) = 4.03, *p* = 0.045, η^2^ = 0.010). To this end, investment in safety empowerment will help enhance the work environment and promote the needed safety climate where an individual will feel supported and enhance their professional service delivery [36]. In addition, employees’ trust in the efficacy of existing safety systems was found to have a weak correlation with management safety justice (*r* (357) = 0.33, *p* < 0.001). In line with this finding, for an organization to encourage the adoption of positive safety culture among its employees, there is the need to apply the right occupational safety and health standards, which in turn requires strong political will and good governance across every stakeholder [37].

Earlier studies have demonstrated a safety climate as highly related to safety participation, and, to promote safety climate in any organization, safety communication, safety training, and safety systems should be actively encouraged within the establishment [38,39,40,41]. From the present study, there was acknowledgement of the role played by communication, training, and having good safety systems as the means of achieving a safety climate within the healthcare sector; however, safety climate was found to be not actively encouraged across the participating organization, as represented in the participants’ responses. Where safety communication and training is neglected, HCWs are likely to harbor mistrust and dissatisfaction with the management, and will most likely engage in unsafe behavior that might compromise either their patient or their personal safety. General comments among the present study participants reflect on this position: “……the management assume we are in the know, considering we are all professionals as such safety training don’t often come up until accident happen”. This statement further demonstrates the need for managers in healthcare organizations to ensure the promotion of policies that reflect their commitment to employee health and safety, especially in LMICs [39]. While the study did not consider the association between safety climate and staff morale, it is, however, worth highlighting the likely association between safety climate in the healthcare establishment and the brain drain of professionals in LMICs like Nigeria. The reasons cited for a high level of exits from these countries’ healthcare systems among professionals include poor safety policies, poor and insecure working conditions, etc., that, in combination, makes work harder to deliver and presents a state of hopelessness among the workforce [42,43].

Previous studies have established a link between safety climate items with occupational accidents, injuries, and illnesses [44,45,46]. Our study does expand on this finding and emphasizes that safety climate items can impact employees’ work ability. Results from questions that measured management effort toward accident prevention include: (1) “Management collects accurate information in accident investigations”; (2) “Fear of sanctions from management discourages employees here from reporting near-miss accidents”; (3) “Management listens carefully to all who have been involved in an accident”; (4) “Management looks for causes, not guilty persons, when an accident occurs”; (5) “Management always blames employees for accidents”; and (6) “Management treats employees involved in an accident fairly”; the mean score range was 2.70 to 2.99. This outcome demonstrates the existence of a fairly low-level safety climate within the participating healthcare facilities, thereby demonstrating the need for further safety culture improvement in order to help reduce the risk of accident and minimize physical and mental impact among the healthcare workers.

## 5. Strengths and Limitations

The present study has both strengths and limitations. A key strength is the ability to adapt the validated NOSACQ-50 questionnaire to measure safety climate among healthcare workers in Nigeria, and, to our knowledge, this is the first study that has considered the application of the survey tool among HCWs in the country. In addition, acceptance by the state ministry of health to approve the study is a demonstration of the hospital management board’s willingness to look into measures that will help enhance the safety climate within healthcare establishments in the region. Furthermore, the present study provides a platform for further application of the survey instruments across the geographical regions of the country.

This study has some limitations. With the proliferation of private healthcare clinics/hospital across the country, a major limitation associated with the study is that the study has failed to ask specific question to enquire if participant works in either a government or private healthcare establishment, which could have helped to compare the safety climates across these two settings. This is recommended in future research work. In addition, because the study only considered HCWs in limited states within the northern part of the country, the present results are limited in terms of generalizability. It can, however, be applicable to other healthcare establishments across the country. Another limitation of the study is the low response rate among the professionals across the region, which was partly due to the data collection approach using an online questionnaire, to which staff with no access to the link were less likely to engage with the survey. In addition, the low research culture in the healthcare establishment, especially in this region, is another likely reason why staff did not engage with the survey.

## 6. Conclusions

The outcome from the study showed a weak level of positive safety climate promotion within the healthcare sector based on the participants’ responses. The results also revealed that, while employees are likely to engage in positive safety behavior, there is the need for more visible leadership commitment to ensure that the interest among the employees is maintained. In order to achieve this goal, enhancement of safety communication, safety training, and the adoption of safety systems at work will encourage HCWs to comply with laid out safety rules and procedures as opposed to voluntary participation in safety practices, which might sometimes not be in the interest of their patients or the organization in general. The study concludes that there is still a long way to go in promoting an effective and positive safety climate in healthcare systems in LMICs, and all actors are encouraged to play an active role to make this a reality. Several challenges that include a lack of political will, weak policies, and limited monitoring by the relevant regulatory body have slowed the pace at which both government-owned and private healthcare establishments promote a safety climate in these countries. To help address this setback, more effort is required at promoting an effective and positive safety climate across the healthcare establishments, including management and healthcare workers’ commitments.

## Figures and Tables

**Table 1 healthcare-09-00661-t001:** Distribution of the mean and Cronbach’s alpha reliability test across the seven dimension using the NOSACQ-50.

Dimension	Items	Mean	SD	Cronbach’s α
1. Management safety priority, commitment, and competence	9	2.9139	0.45780	0.787
2. Management safety empowerment	7	2.7427	0.38206	0.793
3. Management safety justice	6	2.6918	0.47658	0.754
4. Workers’ safety commitment	6	3.0138	0.47543	0.793
5. Workers’ safety priority and risk non-acceptance	7	2.6201	0.44197	0.703
6. Safety communication, learning, and trust in co-workers’ safety competence	8	3.0033	0.41566	0.810
7. Trust in the efficacy of safety systems	7	3.1734	0.43961	0.756
Overall/Total	50	2.8886	0.33431	0.932

**Table 2 healthcare-09-00661-t002:** Description of participants’ socio-demographic characteristics (*n* = 433).

Variable	Frequency	Percentage
Gender		
Male	230	53.7
Female	198	46.3
Total	428	100
Age group		
18–25	18	4.2
26–30	70	16.4
31–35	84	19.6
36–40	72	16.8
41–45	61	14.3
46–50	55	12.9
51–55	53	12.4
56–60	14	3.3
60-above	1	0.2
Total	428	100
Profession		
Medical doctor	63	15.4
Pharmacists	34	8.3
Nurse/Midwife	167	40.7
Lab Scientist/Technician	42	10.2
CHO/EHO/EHT/PHO	65	15.9
Physiotherapist	3	0.7
Others	36	8.8
Total	410	100
Do you have a managerial position?		
No	234	54.7
Yes	194	45.3
Total	428	100

**Table 3 healthcare-09-00661-t003:** One-sample test for the seven domains measured.

Dimensions	t	*df*	*p*-Value	Mean Difference	SD	95% Confidence Interval	
						Lower	Upper
1. Management safety priority, commitment, and competence	129.353	412	0.001 *	2.91391	0.45780	2.8696	2.9582
2. Management safety empowerment	147.469	421	0.001 *	2.74272	0.38206	2.7062	2.7793
3. Management safety justice	115.341	416	0.001 *	2.69185	0.47658	2.6460	2.7377
4. Workers’ safety commitment	130.527	423	0.001 *	3.01376	0.47543	2.9684	3.0591
5. Workers’ safety priority and risk non-acceptance	120.184	410	0.001 *	2.62009	0.44197	2.5772	2.6629
6. Safety communication, learning, and trust in co-workers’ safety competence	147.016	413	0.001 *	3.00332	0.41566	2.9632	3.0435
7. Trust in the efficacy of safety systems	147.232	415	0.001 *	3.17342	0.43961	3.1311	3.2158

* Significant, *p* < 0.001.

**Table 4 healthcare-09-00661-t004:** Summary of the inter-correlation matrix testing the correlation between the seven safety climate dimensions.

Dimensions	M	SD	1	2	3	4	5	6	7
1. Management safety priority, commitment, and competence	2.9200	0.46399	1.000						
2. Management safety empowerment	2.7459	0.39557	0.635	1.000					
3. Management safety justice	2.6923	0.47655	0.632	0.682	1.000				
4. Workers’ safety commitment	3.0201	0.49795	0.477	0.555	0.513	1.000			
5. Workers’ safety priority and risk non-acceptance	2.6226	0.45605	0.464	0.367	0.431	0.385	1.000		
6. Safety communication, learning, and trust, in co-workers’ safety competence	3.0035	0.40956	0.489	0.528	0.471	0.673	0.400	1.000	
7. Trust in the efficacy of safety systems	3.1813	0.43902	0.370	0.375	0.333	0.479	0.354	0.573	1.000

Note: All correlations are significant at *p* < 0.001.

**Table 5 healthcare-09-00661-t005:** Analysis of variance summary for a managerial position measured against the seven dimension considered in the study.

Dimension	*df* (1^#^)	MS	*F*	η^2^
1. Management safety priority, commitment, and competence	406	0.830	3.989 *	0.010
2. Management safety empowerment	415	0.587	4.030 *	0.010
3. Management safety justice	410	2.035	9.190 *	0.22
4. Workers’ safety commitment	417	0.136	0.598	0.001
5. Workers’ safety priority and risk non-acceptance	404	1.386	7.289 *	0.018
6. Safety communication, learning, and trust in co-workers’ safety competence	407	0.360	2.066	0.005
7. Trust in the efficacy of safety systems	409	1.409	7.379 *	0.018

^#^*df* between groups = 1, * *p* < 0.05.

## Data Availability

The data presented in this study are available on request from the corresponding author.

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
