# Peer review of "Assessment of Workplace Safety Climate among Healthcare Workers during the COVID-19 Pandemic in Low and Middle Income Countries: A Case Study of Nigeria"

_healthcare, 2021, doi:10.3390/healthcare9060661_

Round 1

Reviewer 1 Report

Dear Authors,

I appreciate the efforts that went into this paper and I appreciate the opportunity to review it also. This paper "Assessment of Workplace Safety Climate among Health Care Workers during the COVID-19 Pandemic in Low and Middle Income Countries- Case Study of Nigeria" covered a crucial aspect of caregivers during pandemics.

On the other hand, the paper needed some changes I would like to give more specific recommendations for the authors' that I believe will improve the paper. I think this paper will fit well in the journal after the revisions.

INTRODUCTION

It would be necessary to introduce a paragraph that relates ergonomic prevention from safety and to reduce physical and psychological disorders. In general, these aspects would be related to behaviors and cultural climate, among others. For example, mention can be made of other sectors where more work is being done on this aspects and clarified methodology has been applied. I highly recommend citing this publication): https://www.revistadyna.com/search/ergonomic-risk-factors-analysis-with-multi-methodological-approach-assessing-workers-activities-in-b

MATERIALS AND METHODS

Please, include the sample calculation to justify the 445 responses.

DISCUSSION AND CONCLUSION

Please, in this section also relate to the part on the need for ergonomics to mitigate biomechanical and psychosocial risks

REFERENCES

I recommend adding the DOI in articles and webpage in the rest of the cases.

Author Response

Many Thanks for the kind comments made to our paper. 

We have now improved the work and attached our response against each questions/comment raised

Reviewer 2 Report

Revision The summary should be improved in the methodology section, does not identify study objectives and refers to a questionnaire that does not identify whether it is validated or not, taking into account that it is a Nordic questionnaire and the study population is from Nigeria. The results should be described in the summary in a more readable and interpretative manner and leave the many specific data for the content of the article, not in the summary section. There are no conclusions, they should write them to check the coherence between objectives, methodology and results and check if the conclusions are operational.  Regarding the introduction, it should include more current references and although it has found some work safety experiences in sub-Saharan Africa (Uganda and Ghana) it does not talk about the working climate and some are agriculture and not health. There are references of qualitative studies to address working climate and work culture, mixed methodology should be considered for future studies

I would like you to address in the introduction the difference between working climate and work culture, even though in (69-70) you name it, you say you will do it interchangeably, but you are aware that they are not equal concepts.

The paragraph of the introduction (102-108) should be referred to, it is stated something important if theoretical justification

Regarding methodology, very improveable, is disorderly:

It does not identify study objectives

Refered of the instrument before the population

It does not define how it is sampled, it is intuated that it is intentional, but does not explain it, does not speak of the number of participants in this section, explains them in results

The questionnaire says it is validated but we do not know if in the same language and if it is, not in the same culture so validation should be piloted. When you present results, it appears that you are performing an internal validation of it as it speaks of cronbach alpha>0.7, but it is not in that section where to put it.

The presentation of the results is not orderly.We do not know if the results meet the objectives The discussion is improveable.

There is no limitations section of the study.

Finally, the bibliography needs to be improved, there are different fonts and different ways of referencing. You must update the references.

Author Response

Dear Reviewer, 

Many thanks for your comments which did help improve the quality of our initial submission. 

Find attached response to all comments made earlier. 

Many thanks 
